# Validation of Sentinel-3A Based Lake Level over US and Canada

**Karina Nielsen \*** , **Ole Baltazar Andersen** and **Heidi Ranndal**

Geodesy and Earth Observation, The Technical University of Denmark, DK-2800 Kgs. Lyngby, Denmark;
oa@space.dtu.dk (O.B.A.); hvil@space.dtu.dk (H.R.)
\* Correspondence: karni@space.dtu.dk; Tel.: +45-45259764

**Abstract:** Satellite altimetry is an important contributor for measuring the water level of continental water bodies. The technique has been applied for almost three decades. In this period the data quality has increased and the applications have evolved from the study of a few large lakes and rivers, to near global applications at various scales. Products from current satellite altimetry missions should be validated to continuously improve the measurements. Sentinel-3A has been operating since 2016 and is the first mission operating in synthetic aperture radar (SAR) mode globally. Here we evaluate its performance in capturing lake level variations based on a physical and an empirical retracker provided in the official level 2 product. The validation is performed for more than 100 lakes in the United States and Canada where the altimetry based water levels are compared with in situ data. As validation measures we consider the root mean squared error, the Pearson correlation, and the percentage of outliers. For the US sites the median of the RMSE value is 25 cm and 19 cm and the median of the Pearson correlations are 0.86 and 0.93 for the physical and empirical retracker, respectively. The percentage of outliers (median) is 11% for both retrackers. The validations measures are slightly poorer for the Canadian sites; the median RMSE is approximately 5 cm larger, the Pearson correlation 0.1 lower, and the percentage of outliers 5% larger. The poorer performance for the Canadian sites is mainly related to the presence of lake ice in the winter period where the surface elevations are not able to map the surface correctly. The validation measures improve considerably when evaluated for summer data only. For both areas we show that the reconstruction of the water level variations based on the empirical retracker is significantly better compared to that of the physical retracker in terms of the RMSE and the Pearson correlation.

**Keywords:** Sentinel-3A; inland water; validation

## 1. Introduction

Satellite altimetry has been used for inland water applications for nearly three decades proving its value in many studies [1–6]. In the beginning satellite altimetry was mainly applied over a few large lakes and rivers [7,8], but is now an established contributor, routinely providing globally distributed water levels available from services such as the Database for Hydrological Time Series of Inland Waters (DAHITI) [9], Hydroweb [10], and the Global Reservoirs and Lakes Monitor (G-REALM) [11]. Over the years the quality of the data has increased due to improvements of the altimeters, corrections and dedicated processing schemes [12–15]. A substantial leap in the possibilities for inland water applications has been achieved after the launch of CryoSat-2 in 2010 and Sentinel-3A/B in 2016 and 2018, all carrying synthetic aperture radar (SAR) altimeters [16]. The SAR mode has allowed to obtain more accurate water levels for smaller water bodies [17]. The Sentinel-3A and 3B, are operating in SAR mode globally, hence, providing the possibility for a high data quality. A high quality of the data, in terms of precision and accuracy, is an important factor for further use in hydrology for

example, for discharge estimation [18] and to calibrate hydrodynamic models [19]. In addition to errors originating from the altimeter and the corrections (atmospheric and geophysical) used to calculate the water surface elevation, the error of the measurements depends on how well the surface elevation can be defined by the retracker. Compared to the ocean, lakes and rivers are small water bodies and when observed by the altimeter the echo will often be contaminated with signals not necessarily from the nadir water surface. The contamination in the waveform typically originates from the surrounding topography and bright off-nadir targets for example, other water bodies. Noisy waveforms can lead to erroneous surface water elevations, which may degrade the product. To some degree, the degradation depends on the applied retracking algorithm. Hence, evaluating the retrieved water surface elevation is important to continuously improve the products.

Sentinel-3A has now been operating for approximately four years and some studies have already evaluated the data quality for inland water applications. Reference [20] performed validation over small reservoirs in the Ebro basin in Spain against in situ data demonstrating a RMSE between 16 and 28 cm. Reference [14] evaluated Sentinel-3A water levels based on both standard and non standard retrackers at 26 virtual stations at Chinese rivers and demonstrated RMSE values between 12 and 90 cm. The best performance was obtained with a modified version of the Multiple Waveform Persistent Peak (MWaPP) retracker [12]. Reference [21] did several evaluations to quantify the performance of Sentinel-3A and 3B for inland water for example, demonstrated an accuracy of 1.5 cm over lake Issyk-Kyl and showed the consistency of the water surface height between the Sentinel-3A and 3B over Lake Lagoda in Russia.

Here we evaluate the ability of Sentinel-3A to capture water level variations based on the physical Ocean [22] and empirical Offset Centre of Gravity Retracker (OCOG) [23] retrackers available in the official Sentinel-3A altimetry product. The validation is performed against in situ data from more than 100 lakes located in the United States (US) and Canada with an area between 50 and 7000 km$^2$. In this investigation we focus on the overall trend among the considered lakes to provide a general performance estimate for Sentinel-3A. We also quantify, if there is a significant difference in the solutions based on the two retrackers. For selected lakes we investigate the reasons for apparent large deviations between the Sentinel-3A and in situ water levels and discuss the importance and need for alternative quality control when in situ data is not available.

## 2. Material and Methods

### 2.1. Data Sets and Study Area

For this study we consider a total of 138 lakes, 40 lakes in the US and 98 lakes in Canada, with an area between 50 and 7000 km$^2$. The lakes in US and Canada contain 50 and 172 Sentinel-3A crossings, respectively. A satellite crossing over a lake or river is also referred to as a virtual station. These lakes where selected based on the availability of in situ level data. Figure 1 displays the locations of the lakes included in this study.

For validation of the selected lakes We apply in situ level data from the US and Canada, which is freely available from https://maps.waterdata.usgs.gov/mapper/index.html and https://wateroffice.ec.gc.ca, respectively. Figure 1 displays the location of the applied in situ data.

For this study we use the official Level 2 (L2) Sentinel-3A Land altimetry product, in particular, the enhanced measurements in 20 Hz resolution for non time critical purposes available from https://scihub.copernicus.eu/. We consider data from the period March 2016 to May 2020 covering the cycles 1 to 58 from Baselines 3 and 4. The data files provide all the parameters needed to calculate the surface elevation with respect to the EGM2008 geoid model [24].

The Sentinel-3A satellite is a part of the Sentinel family, which is a fleet of missions developed by the European Space Agency (ESA) designed to contribute to the European Commision's Copernicus Program. The Sentinel-3A satellite was launched on 16 February 2016, followed by the launch of the identical Sentinel-3B satellite on 25 April 2018. With the two identical satellites in orbit, a global

coverage is obtained every two days. Each Sentinel-3 satellite flies at an altitude of around 814 km and has a repeat period of 27 days with an orbit similar to that of Envisat to allow for an easier continuation of the time series derived from Envisat and ERS data. As such, the orbit is sun-synchronous and near-polar with a high inclination of 98.65 degrees. The Sentinel-3 satellites have identical orbits but flies 140 degrees out of phase. At the equator there is 52 km between each track when using both satellites. For additional information consult the mission web page https://sentinel.esa.int/web/sentinel/missions/sentinel-3.

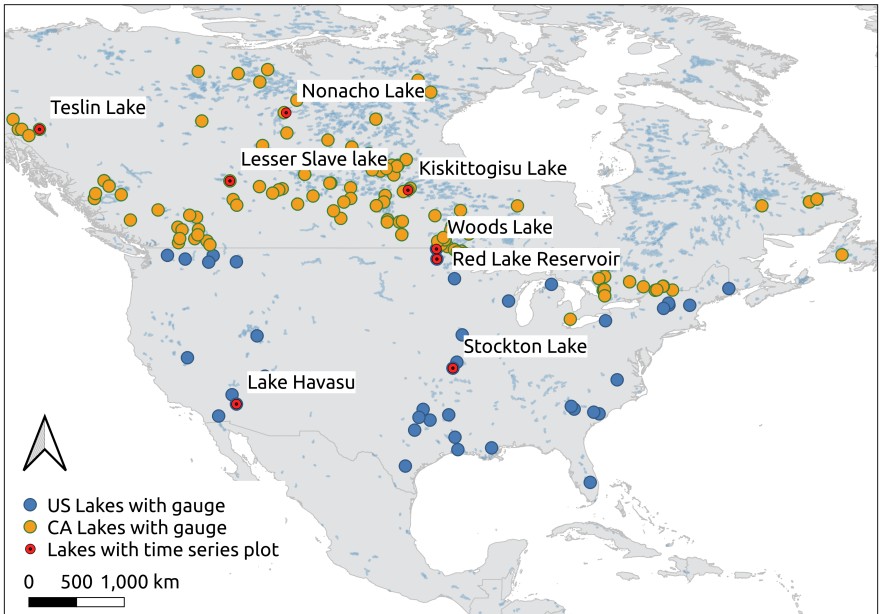

**Figure 1.** Map displaying the location of the in situ level data which is used in the evaluation. The red dots indicates the location of the selected lakes where lake level time series are displayed.

The surface height is measured by the Sentinel-3 Ku/C Radar Altimeter (SRAL). By utilizing the Delay Doppler effect, SRAL has an along-track resolution of approximately 300 m, making it possible to obtain water surface elevations from smaller water bodies than with the conventional Low Resolution Mode (LRM) altimetry missions. The Sentinel-3 satellites are the first to operate in SAR mode on a global scale, whereas the earlier CryoSat-2 mission only operated in SAR mode in selected areas.

Sentinel-3A is operating in an open and closed loop configuration. In the open loop configuration the range window is positioned according to a pseudo elevation model onboard the satellite, while in the close loop configuration the range window is positioned according to where the altimeter senses the surface. Since March 2019 Sentinel-3A has been operating in the open loop tracking mode between 60°N and 60°S. The open loop tracking mode is described in Reference [25] and at the site "Altimeter Open Loop Tracking Command for Hydrology" https://www.altimetry-hydro.eu/. At this sites it is also possible for users to improve future onboard DEMs by adding water surface elevations at S3A/B virtual stations.

In the L2 product, the available Ocean retracker is the SAR Altimetry MOde Studies and Applications (SAMOSA2.5) retracker [22,26]. It is similar to the classical Brown model [27] used for LRM waveforms, but has been developed for SAR altimeter echoes. The SAMOSA retracker assumes a flat underlying surface consisting only of open ocean, that is, without any signals from other surface types. For open ocean waveforms, the SAMOSA retracker is expected to obtain surface elevation estimates of a very high quality. However, for inland water studies, the assumption of a homogeneous surface might result in erroneous surface elevation estimates since the waveforms can be contaminated by off-nadir signals such as calm waters in the lake shores or smaller water bodies adjacent to the lake at nadir.

The other retracker considered in this study is the empirical OCOG retracker [23]. This retracker is similar to the ice-1 retracker, which was applied by Reference [28] to the ERS-1 and -2 and Envisat data to derive ice sheet and land elevations. The OCOG retracker is based on a statistical approach and makes no assumptions about the underlying surface. It simply determines the center of gravity of the waveform, as well as its amplitude and width, from which the retracking point is deduced. The Ice-1/OCOG retracker has previously demonstrated better results for inland waters compared to other LRM retrackers such as the physical Brown ocean model [29–31].

### 2.2. Construction of Sentinel-3A Surface Water Elevations

The altimetry based surface water elevation with respect to a geoid, here EGM2008 [24], is constructed via the follow equation

$$H = h - R_{corr} - N, \tag{1}$$

where $h$ is the altitude of the satellite, $N$ is the geoid height and $R_{corr}$ is the corrected range, that is, the distance between the surface and the satellite, which is constructed as follows

$$R_{corr} = R - R_{atm} - R_{geo} - R_{retrack}, \tag{2}$$

where $R$ is the uncorrected range referenced to the nominal bin, $R_{atm}$ and $R_{geo}$ are atmospheric and geophysical corrections including; wet and dry troposphere, ionosphere, pole tide, and solid Earth tide. $R_{retrack}$ is the retracking correction, which accounts for the difference in range when the actual surface reflection is not represented by the nominal bin in the waveform.

### 2.3. Generation of Sentinel-3A Lake Level Time Series

To extract the Sentinel-3A observations located over the lakes, we use the occurrence product, with a 20% threshold, from the Global Surface Water Explorer [32] and the lake mask from HydroLAKES https://hydrosheds.org/page/hydrolakes [33]. Before estimating the lake level time series we perform a crude outliers filter by removing observations more than 50 m from the lake elevation provided in the HydroLAKES data set. These elevations originate from the EarthEnv-DEM90 digital elevation model [34]. North of 60°N degrees some elevations were replaced by values from the coarser GTOPO30 elevation model.

The lake level time series are automatically reconstructed using the "R" software package "tsHydro" [35], which is available from GitHub https://github.com/cavios/tshydro. In the reconstructing of the water level time series we apply all along-track surface elevation measurements. tsHydro is based on a state-space model, where the process part is a random walk and it is assumed that the observations follow a mixture between Gaussian and Cauchy distributions. This makes the model less sensitive to erroneous observations. Compared to a simple mean or median water level estimate, the state-space model ensures a temporal correlation via the random walk. This implies, that the model assumes that water levels close in time will be more alike compared to water levels that are far from each other in time. Besides the water levels, the model has two model parameters; the standard deviation of the point observation and the standard deviation of the random walk. The water level time series are often referred to as a level 3 (L3) product which hereinafter is applied in the figures.

### 2.4. Validation Measures

As validation measures we calculate a bias corrected root mean squared error (RMSE), the Pearson correlation, and the percentage of outliers. Since the in situ water levels are provided relative to various local vertical references, we remove the median offset between the altimetry and in situ data before calculating the RMSE. The bias corrected RMSE values indicates how well the altimetry represents the

relative water level variations. This is more interesting than representing the absolute water level with respect to a given vertical reference. As an additional validation measure we consider the percentage of outliers. We have here defined an outliers as a measurement that is more than 1 m from the in situ value at the given time. The value of 1 m is arbitrary and chosen only with the purpose to have a fixed measure. It is here important to emphasize, that we do not remove these outliers when constructing the water level time series. The validation measures are calculated for each virtual station.

Here we aim at making a general statement about the performance of the Sentinel-3A satellite in terms of reconstructing water level variations in lakes. We therefore focus on the median of the validation measures for all virtual station instead of the individual cases. The median is applied since the distributions of the validation measures are not symmetric. To quantify the uncertainty of the summary measures and to test if there is a significant difference between the Ocean and OCOG based results, we use a bootstrap approach, since the distribution of the validation measures are not Gaussian. In Bootstrapping a large number of equally likely data sets are created by sampling with replacement from the original data. The method relies on independent observations and that the data sample is large enough to represent its distribution. In this study we assume that the validation measures are independent among the lakes, but correlated within a given lake if more that one sentinel-3A track is present. Hence, when creating the bootstrap data sets we sample among the lakes instead of the tracks. We create 1000 bootstrap data sets, which provides a distribution of the median from which a confident interval can be obtained. To test if the validation measures based on the two L2 observations (Ocean and OCOG) are significantly different we test if zero is within the 95% confidence when considering the pairwise differences.

## 3. Results

The results of the validation in terms of a bias corrected RMSE, the Pearson correlation, and the number of outliers is performed at 50 virtual stations distributed over 40 lakes in the US and 172 virtual stations in Canada distributed over 98 lakes are shown in Figures 2 and 3.

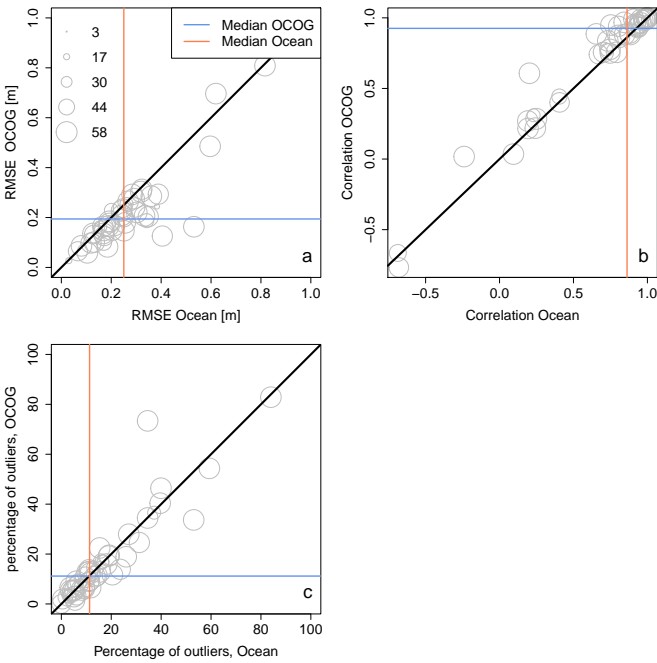

**Figure 2.** Scatter plot of (**a**) root mean squared error (RMSE) values, (**b**) Pearson correlation, and (**c**) percentage of outliers for the US lakes. The vertical orange and blue lines represent the median RMSE values for the Ocean and Offset Centre of Gravity Retracker (OCOG) retrackers, respectively. The size of the circles indicates the number of cycles used in the calculation of the summary measures. In order to detail the findings, only RMSE values below 1 m are shown.

In the subplots of Figures 2 and 3 the Ocean and OCOG based results are plotted against each other and the size of the circles indicates the number of cycles (out of a maximum of 58) used to calculate the validation measures. To highlight the details of the RMSE values (Figures 2a and 3a,b), the range is limited to 1 m, but for both areas a handful of sites have an RMSE value larger than 1 m. The orange and blue lines indicate the median RMSE value for the Ocean and OCOG based results, respectively. The medians of the validation measure distributions and their 95% confidence interval are provided in Table 1. The RMSE values for the US sites are generally below 0.5 m, the Pearson correlation is generally above 0.75, and the percentage of outliers is generally below 20% (Figure 2).

**Table 1.** The median and 95% confidence interval of the validation measures; RMSE, Pearson correlation and the percentage of outliers, for the two retrackers Ocean and OCOG. The subscript *s* indicates solution based on summer data only.

| | RMSE [m] | 95% Conf | Cor | 95% Conf | % Outliers | 95% Conf | # Cycles |
|---|---|---|---|---|---|---|---|
| US ocean | 0.25 | [0.19 0.31] | 0.86 | [0.75 0.94] | 11.3 | [8.2 15.5] | 56 |
| US OCOG | 0.19 | [0.16 0.23] | 0.93 | [0.84 0.97] | 11.2 | [7.7 14.5] | 57 |
| CA ocean | 0.32 | [0.28 0.36] | 0.77 | [0.72 0.85] | 14.7 | [13.6 16.6] | 38 |
| CA OCOG | 0.25 | [0.20 0.30] | 0.84 | [0.76 0.90] | 17.2 | [15.1 18.8] | 38 |
| CA ocean$_S$ | 0.12 | [0.11 0.14] | 0.87 | [0.83 0.93] | - | - | 17 |
| CA OCOG$_S$ | 0.09 | [0.08 0.10] | 0.95 | [0.91 0.97] | - | - | 17 |

Figure 4 shows some examples of water level time series, based on the OCOG based water levels, for the US lakes. In most cases the lake level variations are well reconstructed by the Sentinel-3A satellite, an example of this is Stockton Lake (Figure 4a). For this lake the RMSE, Correlation, and percentage of outliers is 8 cm, 0.99, and 5%, respectively. For lakes with a high frequency in the water level variation, such as Lake Havasu, the low frequency change is mostly captured (Figure 4d). The validation measures for this lake are; 18 cm (RMSE), 0.75 (correlation), and 2% (outliers). In a few cases, such as Red Lake Reservoir (Figure 4b), the RMSE value is above 2 m. This is mainly due to the open loop mode mask change in March 2019, where the water level become wrongly estimated after this update (Figure 4b). The validation measures for Red Lake Reservoir are; 7 m (RMSE), −0.77 (correlation), and 9% (outliers). For some lakes, located at high latitude, such as Woods Lake, Red Lake Reservoir, Mille Lac, and Mullet Lake there is a relative large deviation between the altimetry and the in situ based lake levels in the winter periods (Figure 4c). The validation measures for Woods Lake are; 29 cm (RMSE), 0.76 (correlation), and 7% (outliers).

The RMSE values for the Canadian sites are generally below 0.7 m, the Pearson correlation is generally above 0.6, and the percentage of outliers is generally below 30% (see Figure 3). For the Canadian sites the median of the RMSE values is 6–7 cm larger, the correlation is approximately 0.1 lower, and the percentage of outliers is approximately 5% larger compared to the US lakes (Table 1).

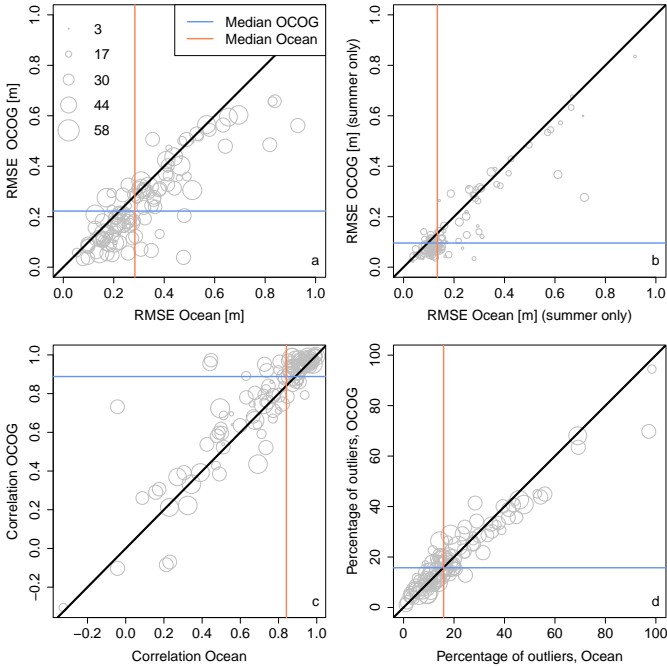

**Figure 3.** Scatter plot of (**a**) RMSE values, (**c**) Pearson correlation, and (**d**) percentage of outliers for the Canadian lakes. The vertical orange and blue lines represent the median RMSE values for the Ocean and OCOG retrackers, respectively. The size of the circles indicates the number of cycles used in the calculation of the summary measures. To allow a greater detail, only RMSE values below 1 m are shown. Subplot (**b**) shows the RMSE values based on summer data only.

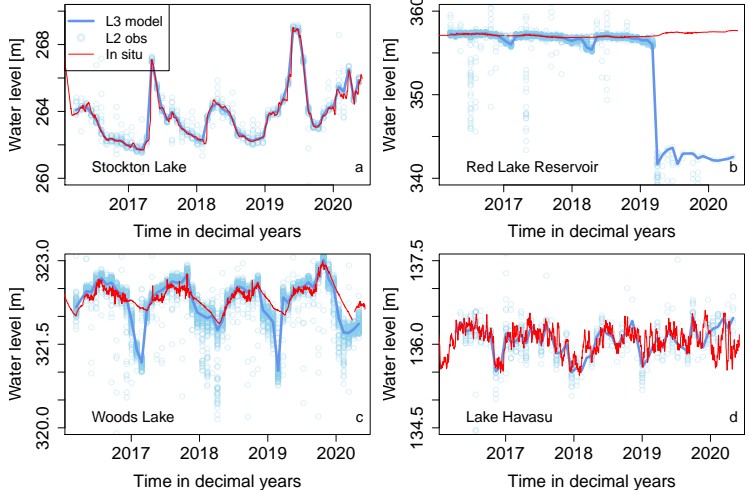

**Figure 4.** Lake level time series for selected US lakes (**a**) Stockton Lake, (**b**) Red Lake Reservoir, (**c**) Woods Lake, and (**d**) Lake Havasu.

Figure 5 displays two typical lake level time series from the Canadian sites. Lake Teslin, located in the Cassiar Mountains (see location in Figure 1), has a quick water level rise in the spring. This lake level signature is also observed for several other lakes in the same area. Lesser Slave Lake, located on the Canadian Prairie (see location in Figure 1), has an annual water level variation of approximately a meter. As for Woods Lake in the US the Sentinel-3A based lake levels are too low in the winter periods. This lake level signature is observed for several other lakes on the prairie. To evaluate the influence of the poor surface elevation estimates in the winter period we calculate the validation measures based only on summer data (measurement in the period 1. June to 1. November). This results in a reduction of approximately 20 cm for the median RMSE value and an increase of 0.1 in correlation (Table 1).

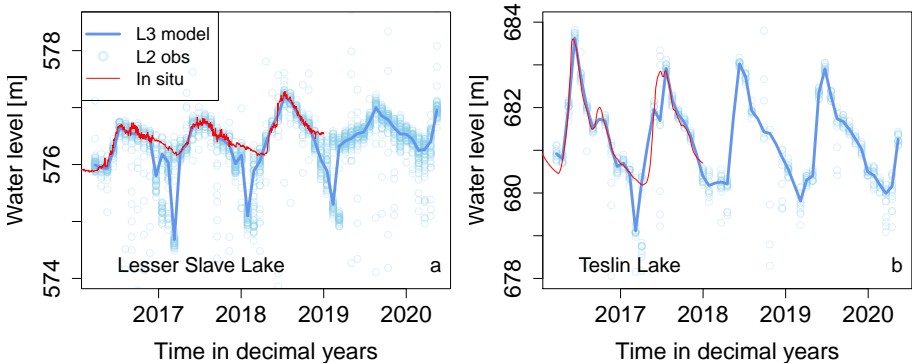

**Figure 5.** Typical lake level time series for Canadian lakes (**a**) Lesser Slave Lake and (**b**) Teslin Lake.

In a few cases, for example, for Lake Kiskittogisu and Nonacho (Figure 6), we find an RMSE value above 2 m. For Lake Kiskittogisu (track number 317) the water level deviates from the in situ data by a few meters in some cycles, which leads to an incorrectly reconstruction of the water level time series. For Lake Nonacho (track number 360) the water level measurements span a large elevation range without a clear indication of a distinct surface leading to a complete failure of the time series model. As a reference Figure 6 also shows a track for the same lakes where the water level time series is correctly reconstructed.

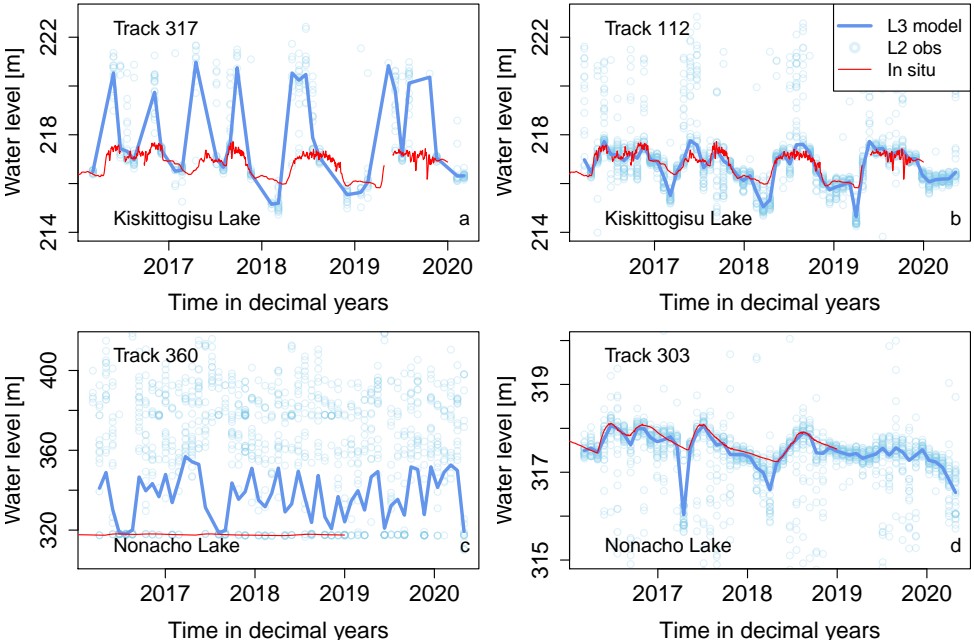

**Figure 6.** Examples of two lakes, where the time series model fails to reconstruct the observed water level due to noisy data. Subplots (**a**,**b**) show lake level time series for lake Kiskittogisu based on track number 317 and 112, respectively. Subplots (**c**,**d**) show lake level time series for lake Nonacho based on track number 360 and 303, respectively.

Generally, there is an indication of a better performance of the OCOG based results (Table 1), where the RMSE value is lower and the correlation is higher. To quantify if this difference is significant we estimate the 95% confidence interval for the medians of the pairwise differences of the three validation measures. The result is shown in Table 2. We find that the OCOG based solutions are significantly better in terms of the median RMSE and correlation (since zero is not included in the confidence interval), whereas it is not significantly better in terms of the percentage of outliers.

The validation measures for all lakes are included in an electronic supplementary. This also includes plots of all the water level time series and the time series themselves for both retrackers.

**Table 2.** The median and 95% confidence interval of the difference (ocean—OCOG) in the validation measures; RMSE, Pearson correlation and the percentage of outliers, between the two retrackers ocean and OCOG. The subscript *s* indicates the case based on summer data only.

|        | Δ RMSE [m] | 95% Conf | Δ Cor | 95% Conf | Δ% Outliers | 95% Conf | # Cycles |
|--------|-----------|----------------|--------|-------------------|-------------|------------------|----------|
| US     | 0.021     | [0.008 0.046]  | −0.013 | [−0.036 −0.003]   | −0.001      | [−0.007 0.021]   | 57       |
| CA     | 0.053     | [0.045 0.066]  | −0.027 | [−0.041 −0.014]   | −0.007      | [−0.018 0.003]   | 38       |
| CA$_S$ | 0.026     | [0.020 0.037]  | −0.023 | [−0.028 −0.015]   | -           | -                | 17       |

## 4. Discussion and Conclusions

In this study, we have evaluated the performance of Sentinel-3A in capturing lake level variation based on a comparison of more than 100 lakes with in situ data. The performance was quantified by the three validation measures a bias corrected RMSE, the Pearson correlation, and the percentage of outliers. The medians of the validation measures are 19 and 25 cm (RMSE), 0.93 and 0.84 (correlation), and 11% and 17% for the US and Canadian sites, respectively. Hence, if these lakes are considered representative for lake in general, we can expect a performance with these validation measures or a better for a randomly selected lake in 50% of the cases. We have also demonstrated that the OCOG based solution is significantly better in terms of the RMSE and correlation. Hence, for a randomly selected lake we will obtain a 2 to 5 cm smaller RMSE and a 0.01 to 0.03 increase in the correlation or better in 50% of the cases when applying the OCOG based surface elevations compared to the Ocean based. In terms of the percentage of outliers the difference is not significant.

The median RMSE calculated from the individual virtual station based RMSE values for the Canadian lakes is approximately 5 cm larger and the correlation is 0.1 lower than for the US lakes. This is mainly caused by a deviation between the gauge level and the altimetry based levels in the winter periods where both retrackers provide surface elevations that are too low. This deviation is presumably related to lake ice which may cause a reflection from both the ice-air and ice water interface. Both retrackers tend to capture a surface elevation which presumably is related to the ice-water interface. Some studies have investigated this issue either to determine the ice thickness from altimetry [36] or to improve the processing to obtain valid winter water levels [37]. However, based on this extensive validation of the Canadian and also the US lakes, the winter lake level measurements in the official Sentinel-3A product may not be valid in case of lake ice cover. The degrading effect of lake ice on the validation measures is quantified by calculating the validation measures for the Canadian lakes based on summer data only which then improves considerable.

For the Canadian lake, Nonacho, several of the measurements from track number 360 are 40 or more meters away from the nadir water surface indicating that the range window is not correctly positioned. The left subplot of Figure 7 shows the TanDEM-X elevation model (https://tandemx-science.dlr.de/) in the vicinity of this lake. The topography in this area is changing quickly and several smaller lakes are located at different elevations in the vicinity of Lake Nonacho. This might explain why the range window is not always adjusted correctly. For the Canadian lake, Kiskittogisu, the water level measurements are just a few meters off, so in this case the error is most likely related to the retracking procedure. Depending on how large a water level amplitude we expect an error in the water level of just a few meters might not be detected if in situ data is not available. The right subplot of Figure 7 shows the local elevation variations and the location of the Sentinel-3A tracks in the vicinity of Lake Kiskittogisu. The topography around the lake is quite smooth, but the location of the Sentinel-3A track, number 317, is close to the lake shore which might explain the poor surface elevations for some cycles of this track.

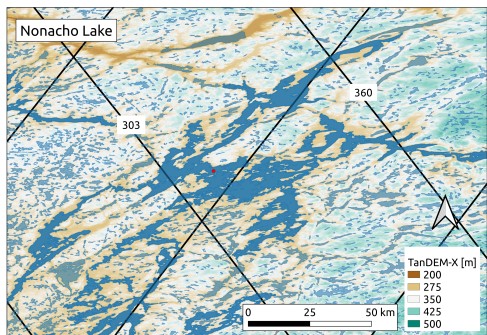 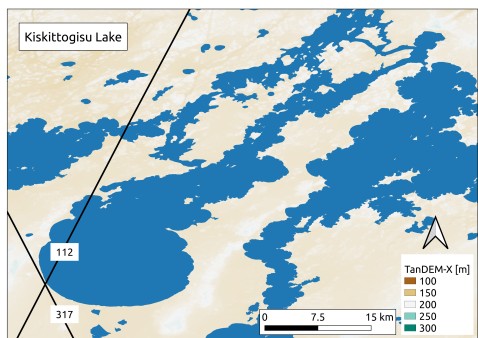

**Figure 7.** Maps displaying the local topography in the vicinity of Lake Nonacho (**left**) and Lake Kiskittogisu (**right**). The Elevations are WGS84 ellipsoid heights from the TanDEM-X elevation model.

The benefits of water levels obtained from radar altimetry missions, are the global and political unconstrained coverage. However, to make decisions or draw conclusions based on altimetry, indicators of whether a given measurement or a water level time series can be trusted is needed. This would greatly strengthen the use of altimetry data for inland water applications. To test, if we objectively can identify the erroneous measurements we have analyzed the waveform parameters; maximum power, pulse peakiness, and the OCOG width and since, the applied time series model provides a valid reconstruction of the lake levels for the majority of the lakes, we can potentially use the model fit to evaluate the validity of the reconstructed time series. For Lake Nonacho we find that the maximum power [counts] generally is lower when the range window is not correctly adjusted. The OCOG width is larger and the pulse peakiness is lower (Figures A1 and A3). For Lake Kiskittogisu, this distinction is not visible (Figures A2 and A4 in Appendix A). For Lake Nonacho the standard deviation of a point observation is estimated by the model to 14 m and 21 cm for track 360 and 303, respectively and for Lake Kiskittogisu the estimates are 40 cm and 27 cm for track 317 and 112, respectively. Hence, again, the distinction is more clear for Lake Nonacho. To understand if it is possible to make some general assumptions based on the waveform parameters more lakes need to be examined and more research on this topic is necessary. Another approach to potentially enhance the confidence in the reconstructed lake level time series would be to include all tracks and measurements from other altimetry missions when reconstructing the lake level time series.

Lake Nonacho, located at a latitude of 61.71°N, is currently mapped by Sentinel-3A in the closed loop tracking mode. In upcoming versions of the open loop mask it is planned to include land surfaces (not the ice sheets) above 60°N. Hence, potentially, the range window will be correctly positioned for Lake Nonacho, which will increase the number of valid surface elevation measurements. Canada is important in terms of lakes, since, according to the HydroLAKES database, more than 0.7 million lakes are located in Canada. Hence, the quality of the future on-board DEM is important to capture the lake water level from as many lakes as possible. As shown in the left part of Figure 7 the topography can have relatively large variations over short distances. Hence, fixing the range window to capture, for example, Lake Nonacho, might result in missing the signal from neighboring lakes at a different elevation, which is the case for the Red Lake reservoir (Figure 4).

We have here focused on the overall performance of Sentinel-3A and for this reason tough interesting and important, we have not investigated and quantified the influence of various factors on the validation measures such as topography, the open/closed loop mode mask, the length of the crossing and the location of the track with respect to the lake. However, we have investigated if there is a relation between the lake area and the RMSE, as one would expect that the smaller lakes would have a larger RMSE since their surface elevation measurements are more prone to land contamination due to their size. We find that this is true for a few smaller lakes but not the general case (see Figure A5 in Appendix A). Hence, as expected, the lake area does not seem to have a dominant influence on the ability of Sentinel-3A to correctly capture lake level variations. What seems to have an influence is

factors such as lake ice, the surroundings for example, topography in the vicinity, and shape of the lake shores.

To conclude, we find that Sentinel-3A, in most cases, is able to map the lake level variations for the lakes considered in the study. Large deviations from in situ data is observed if for example, the range window is wrongly positioned, unfortunate updates in the open loop mode mask, and the presence of lake ice. We find that the empirical retracker, OCOG, is significantly better than the physical retracker, Ocean, in capturing the water level variations terms of the validation measures RMSE and the Pearson correlation.

**Supplementary Materials:** The following are available online at http://www.mdpi.com/2072-4292/12/17/2835/s1.

**Author Contributions:** K.N. designed and performed the data analysis. K.N. wrote the majority of the paper and H.R. has contributed considerably to Section 2.1. K.N., O.B.A., and H.R. contributed to the discussion of the results and revised the paper. All authors have read and agreed to the submitted version of the manuscript.

**Funding:** This research was partly funded by the ESA project WorldWater grant number AO/1-9679/19/I-DT.

**Acknowledgments:** The authors would like to acknowledge Anders Nielsen for statistical guidance and the anonymous reviewers for their constructive comment and suggestions which has improved this manuscript. The authors would also like to thank ESA for freely distributing the Sentinel-3A data.

**Conflicts of Interest:** The authors declare no conflict of interest.

## Appendix A

Figures A1 and A2 display waveform examples from the lakes Nonacho and Kiskittogisu, respectively. For Lake Nonacho the waveform power is lower for the erroneous measurements (surface elevations above 318 m). For Lake Kiskittogisu waveforms related to erroneous measurements (surface elevations above 218 m) have two peaks which are very close.

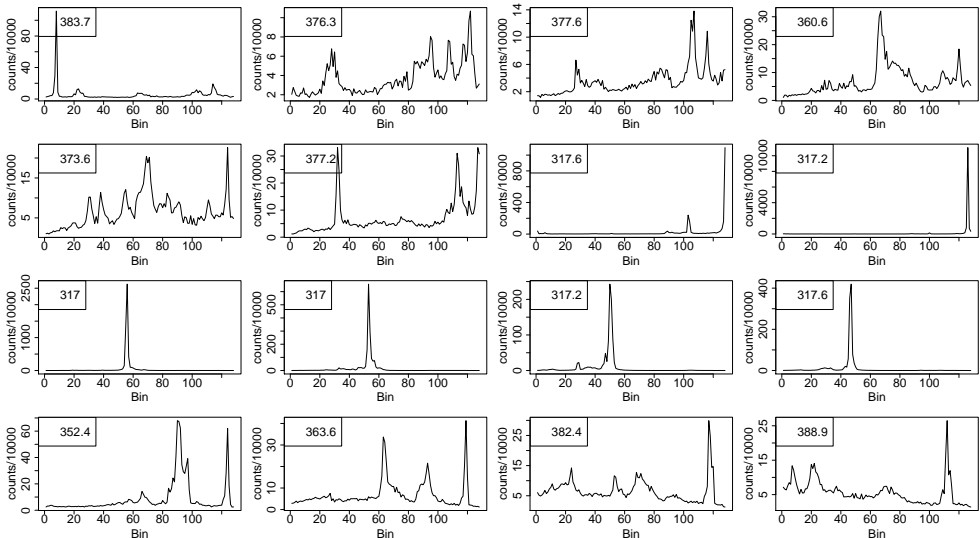

**Figure A1.** Waveform examples for the Canadian Lake Nonacho. The legend displays the retracked surface elevation.

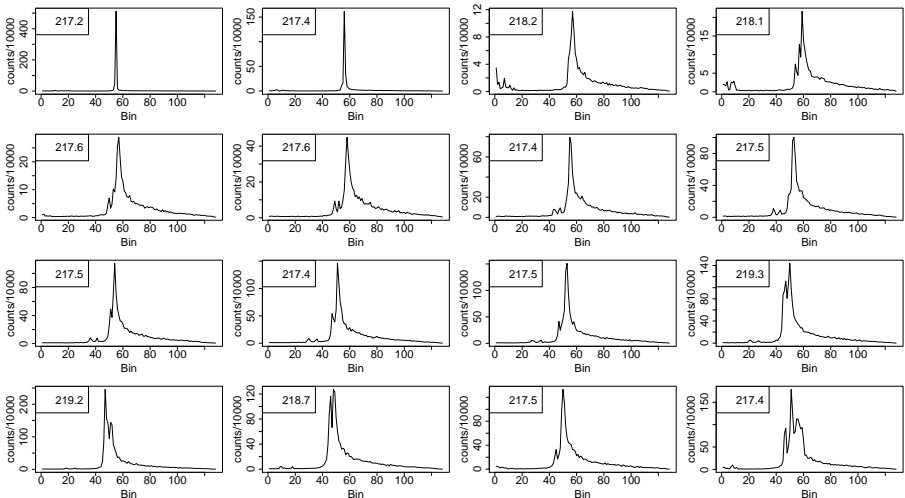

**Figure A2.** Waveform examples for the Canadian Lake Kiskittogisu. The legend displays the retracked surface elevation.

Figures A3 and A4 display the waveform parameters; maximum power, pulse peakiness, and OCOG width as a function of cycle for the lakes Nonacho and Kiskittogisu. The red and blue colors represent measurements with an assumed incorrect and correct surface elevation, respectively.

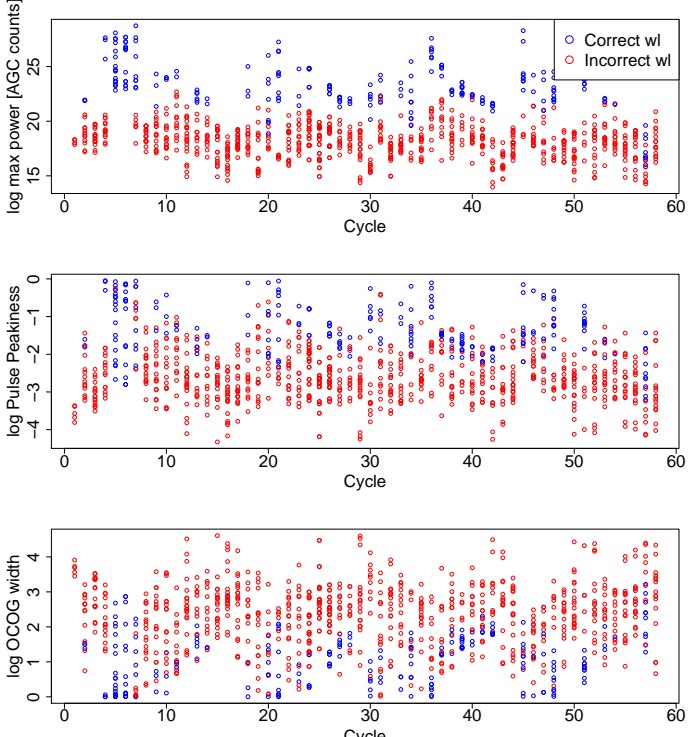

**Figure A3.** Waveform parameters; maximun power, pulse peakiness, and OCOG width as a function cycle for Lake Nonacho track number 360. Measurements are displayed in blue if the surface elevation agrees with in situ data and in red if not.

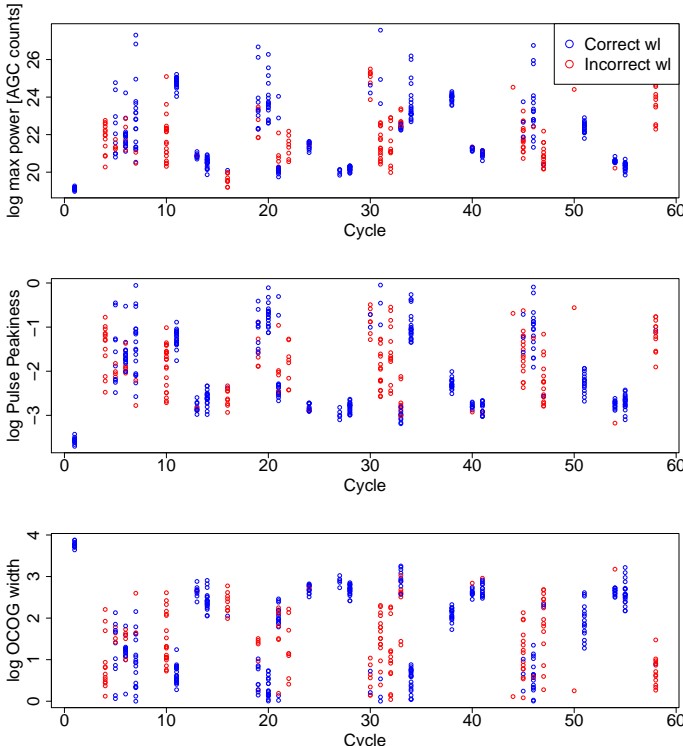

**Figure A4.** Waveform parameters; maximum power, pulse peakiness, and OCOG width as a function cycle for Lake Kiskittogisu track number 317. Measurements are displayed in blue if the surface elevation agrees with in situ data and in red if not.

Figure A5 displays the relation between the RMSE and lake area for the lakes considered in this study.

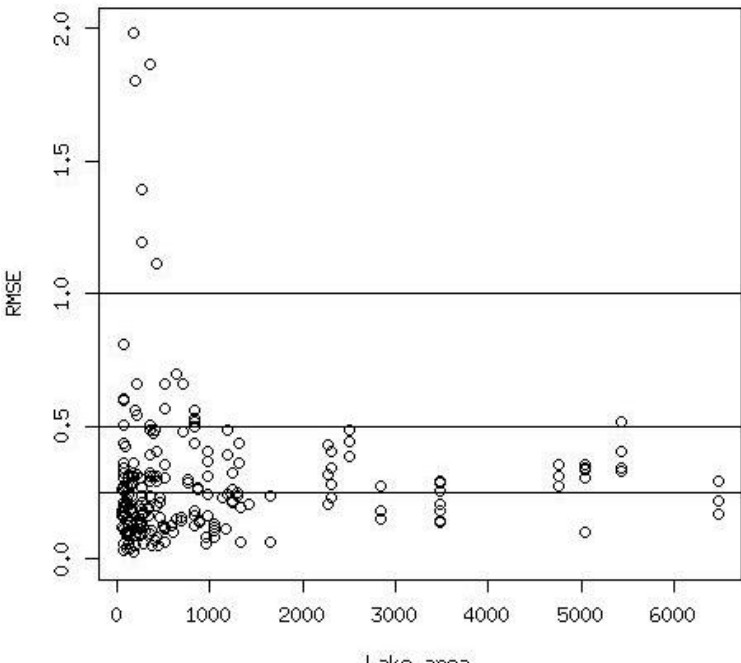

**Figure A5.** Scatter plot of the RMSE as a function of lake area. The function of the horizontal lines is only to make the relation more clear.

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
