# Peer review of "Validation of Sentinel-3A Based Lake Level over US and Canada"

_remotesensing, doi:10.3390/rs12172835_

Round 1
Reviewer 1 Report
This subject is very interesting and important for users and I am glad this paper has been written. This study raises an interesting problem focusing on these two retrackers. The comparison is relevant as they are the only retrackers hydrology users have access to in the PDGS Land L2 products, and users are confused which they should use, why and when.
However, the present study presents raw results of a comparison between two very different retracking algorithms (one is physical designed for Brownian echoes, and the other one is empirical, designed for ice-covered ocean). We know that none of these retrackers are designed for the many situations we face in hydrology. However, this is what users and the mentioned services have. Though the subject is interesting, the study merely scratches the surface and lacks, in my opinion, in-depth analyses and discussions. To synthetize my comments, this study proves that the OCOG-derived measurements are better edited with the TsHydro algorithm, not that the OCOG measurements are more accurate than the SAMOSA (Ocean) ones. Users need to understand why in order to adapt it to their framework.

Author Response
Dear Reviewer 1,
Thank you for the fruitful and constructive comments and suggestions. We hope that you will find our answers satisfactory.
On behalf of all authors
Best regards,
Karina Nielsen

Reviewer 2 Report
Validation of lake water level data of satellite altimetry is very important to its inland water applications. The inland water level of Sentinel-3A are validated by in situ data of US and Canada. The work of this paper is help for the Sentinel-3A data application on the continental water bodies. Some revisions are needed as following.
- Line 58, ‘insitu’ is wrong, it is ‘in situ’.
- add more details about the introduction of Sentinel-3A SRAL and its data.
- Line 72, ‘R is the distance between the satellite and the surface’ is not need here. There is no R in equation (1).
- Line 73-74, Ratm and Rgeo are not clear, please rewrite the sentence.
- Line 101, ‘i March 2019’ should be ‘in March 2019.
- Line 105-106 and Figure 3c, why the deviation is large in winter. Please analyze and give some possible reasons.
- the analyses and the average results are given in section 3 by the comparisons of all lakes without considering the area of the lake. The waveforms of altimeter are influenced by the area of lake and the location of track in the lake, so the precision of lake water level of altimeter is related to the area of lake and the location of the track. The comparisons and RMSE in the bin of lake area should be analyzed to study the precisions of water level data of altimeter for the lakes with different areas.
- the names of lake in Figure 5a/5b ‘Kiskittogisu’ are not the same as that in other places.
- different tacks of the same lake give different results in Figure 5. Give the figure of the location of tracks in the lake area of KiskiKiskittogis as Figure 6 to analyze the reason of the difference.
- Line 133, this sentence is not clear.
- Line 193, ‘A2’ should be ‘A3’.
12 Line 197, ‘A3’ should be ‘A2’.
Author Response
Dear Reviewer 2,
Thank you for the fruitful and constructive comments and suggestions. We hope that you will find our answers satisfactory.
On behalf of all authors
Best regards,
Karina Nielsen

Reviewer 3 Report
Sentinel-3A can detect the level of the lake. The paper selects more than 100 lakes in the United States and Canada to evaluate the detection performance of Sentinel-3A. The results and conclusions are useful. However some elaboration is required.
1. In Fig. 2, it is not appropriate to use frequency in the ordinate. It is more appropriate to indicate the total number of samples and the number of samples corresponding to each RMSE.
2. There are three lines in Figure 3, which need to be explained in the paper. In particular, the relationship between solid blue lines and dotted lines.
3. Error analysis should include correlation coefficient, absolute error, relative error, etc. it is suggested to add these items.
4. It can be seen from Fig. 3 and Fig. 4 that the absolute measurement error of samples is still very large. It is suggested to focus on analyzing and discussing the causes of these errors.
Author Response
Dear Reviewer 3,
Thank you for the fruitful and constructive comments and suggestions. We hope that you will find our answers satisfactory.
On behalf of all authors
Best regards,
Karina Nielsen

Reviewer 4 Report
1st Revision
Title Validation of Sentinel-3A based lake level over US and Canada
General comments
The manuscript displays the evaluation of Sentinel-3A to monitor lake levels in USA and Canada. Using the “R” software package “tsHydro”, the authors constructed the Sentinel-3A time series using two retrackers (OCOG and Ocean retracker) and they were validated with in-situ observations. The results are promising, the obtained Root Mean Square Error (RMSE) values are similar to studies in other areas. Furthermore, the authors discussed the possible sources of error for the high RMSE obtained in some lakes. The work is interesting and encourages the use of altimeters with Synthetic Aperture Radar (SAR) in monitoring lakes.
In general, the manuscript is well-organized and well-written, it’s easy to read. However, there are a few comments and corrections that I suggest the authors to address before publication. Thus, my recommendation is minor revision.
Specific comments
My main three comments/questions are the following:
- At least for me, from Section 2.3, it is not clear how the Sentinel-3A time series are constructed. More specifically, I would like to know if the track points selected are the closest one to the in-situ observations or it is used all the track points available over the lake.
- Is the area of the lake a factor to take in account in the RMSE analysis regardless the retracker used?
- Did the authors test how the comparison between Sentinel-3A and in-situ date change if a low-pass filter is applied?
Line 8: add a space between “lakes” and “in”
Line 66: add “for” after “data”
Line 118: correct the name of the lake Kiskittogisu.
Figure 5: In the figure caption, there is a typo error in the name of the lake shown in panel a) and b).
Line 146: correct the name of the lake Kiskittogisu.
Line 160: correct the name of the lake Kiskittogisu.
Line 162: correct the name of the lake Kiskittogisu.
Line 179: a typo error in values
Line 181: replace “a the” with “to the”
Line 183: replace “a” with “as” after “low”
Appendix: check the spelling of Lake Kiskittogisu.
Author Response
Dear Reviewer 4,
Thank you for the fruitful and constructive comments and suggestions. We hope that you will find our answers satisfactory.
On behalf of all authors
Best regards,
Karina Nielsen
